# Biodegradable Nonwoven Materials with Antipathogenic Layer

Longina Madej-Kiełbik [1],*, Karolina Gzyra-Jagieła [1,2], Jagoda Jóźwik-Pruska [1], Maria Wiśniewskia-Wrona [1] and Marzena Dymel [1]

[1] Lukasiewicz Research Network—Lodz Institute of Technology, 19/27 M. Sklodowskiej-Curie Street, 90-570 Lodz, Poland; karolina.gzyra-jagiela@lit.lukasiewicz.gov.pl (K.G.-J.); jagoda.jozwik-pruska@lit.lukasiewicz.gov.pl (J.J.-P.); maria.wisniewska-wrona@lit.lukasiewicz.gov.pl (M.W.-W.); marzena.dymel@lit.lukasiewicz.gov.pl (M.D.)

[2] Faculty of Material Technologies and Textile Design, Lodz University of Technology, 116 Żeromskiego Street, 90-924 Lodz, Poland

* Correspondence: longina.madej-kielbik@lit.lukasiewicz.gov.pl; Tel.: +48-42-307-2170

**Abstract:** Biopolymer composites have received increasing attention for their beneficial properties such as being biodegradable and having less influence to the environment. Biodegradability of materials has become a desired feature due to the growing problems connected with waste management. The aim of the paper is to emphasize the importance of biodegradable textile materials, especially nonwoven materials with an anti-pathogenic layer. The article refers to the definitions of biodegradation, degradation and composting processes, as well as presenting methods of testing biodegradability depending on the type of material. The study gives examples of biodegradation of textiles and presents examples of qualitative and quantitative methods used for testing antimicrobial activity of biodegradable nonwovens with an anti-pathogenic layer.

**Keywords:** biodegradable nonwoven materials; biodegradable process; antipathogenic layers; antimicrobial activity; circular economy; protection of the environment





## 1. Introduction

Nonwoven industries have grown significantly and now present a broad array of engineered-fiber and polymer-based products. It continues to be adaptive and creative, and has a great potential for innovation. The real challenge for the nonwovens industry where large volume production also means large volumes of waste, is the search for biodegradable polymers. Reuse of materials is a major area of interest for many companies. Activities will focus on finding new applications for such waste or complete reuse of the used product. Therefore, the use of biodegradable biopolymers such as polylactide (PLA), polyhydroxyalkanoate (PHA), polyhydroxybutyrate (PHB) and mixtures thereof has become increasingly popular in recent years. It is likely that a generation of new biodegradable products will replace the existing state-of-the-art. These new materials have to be economically sustainable—today, the nonwovens industry has many sustainable solutions that are not yet economically profitable. However, niche markets are open for new biofriendly products.

The nonwoven industry has adapted technologies from pulp and paper and extrusion industries to create the desired products at a reasonable cost. Automated conversion has been a major part of the nonwoven supply chain for many years. Nonwovens belong to one of the fastest growing textile sectors. They are flat, porous plastics made by mechanical, thermal, or chemical entangling of fibers [1]. These nonwovens can be produced from synthetic and natural fibers alike or directly from polymers/biopolymers by a diversity of techniques that involve web creation and bonding. Non-wovens are often used as single use or short-life products, leading to disposability related problems [2].

In recent years, the demand for biodegradable materials has increased worldwide, especially in the area of disposable products. Currently, the markets for biodegradable

products include a wide range of products such as packaging materials (trash bags, wrappings, food containers) [3], disposable nonwovens or hygiene products (diapers, disposable underlays, swabs), disposable tableware (containers, egg cartons, toys) and agricultural articles [4]. One of the key requirements in designing and developing new products is being environmentally conscious [3]. The solution in this regard is biodegradable or compostable nonwovens which can support the circular economy [5].

More and more emphasis is placed on the improvement and use of clean technologies in order to reduce the consumption of resources and reduce the level of pollution, in support of sustainable development [6]. Nonwovens can be long-lasting or have a short lifecycle. Most nonwovens are single-use products, which are designed for specific applications. An important group of nonwovens is materials with antiviral and antibacterial properties. The dominant nonwoven available in the market is polypropylene (PP)-based, which, however, is nonbiodegradable and partially responsible for the current burden of plastic pollution [7–11] The PP fragments are bio-fragmented into microplastics, leading to their distribution in soil ecosystems [12,13].

There are many methods to improve the antipathogenic properties of nonwovens, such as coating with salt [14,15] and metal nanoparticles [16–18], and the incorporation of polysaccharides [19]. Despite advances in the production of nonwovens with biocidal properties, the main problem is that most of the modifications are performed on non-biodegradable PP-based nonwovens, which create significant environmental concerns after disposal [20]. Therefore, the biodegradable nonwovens with functions similar to PP or better are especially engaging.

To solve the issues related to environmental problems, developing bio-based or compostable nonwoven fabrics endowed with bactericidal and virucidal activity is extremely important. The next important step is to determine the antibacterial activity. There are many methods to assess the antimicrobial activity of nonwovens. Some of them will be discussed in this article. As can be seen in the literature, polymers with antimicrobial activity are known; for example, a guanidine-based polymer has been favorably received due to its strong antibacterial and antiviral properties with high biocompatibility. This makes it a promising application in various areas such as medicine, packaging, food, and agriculture [21–23]. Many of the polymers can also be covalently grafted on different substrates.

During the SARS-CoV-2 pandemic, a lot of materials have been tested for the production of antipathogenic face masks including silver, [24,25] copper and oxides of copper, [26,27] titanium dioxide, [28] and graphene. [17,29]. Nanomaterials of noble metals have reaped much interest for their antibacterial activity [30]. Antimicrobial layers have attracted attention due to high demands in the health field, to lessen the risk of infection by various bacteria or viruses. Bacterial or viral infections are a significant threat to human health today. Nonwoven materials containing anti-pathogenic layers can be helpful in this regard.

The aim of the paper is to emphasize the importance of biodegradable textile materials, especially nonwoven materials, with an anti-pathogenic layer. The article refers to the definitions of biodegradation, degradation, and composting processes. It also discusses exemplary methods of testing the anti-pathogenic properties of nonwovens, and emphasizes the importance of biodegradation.

## 2. Technologies of Producing Biodegradable Nonwovens

Nonwovens are defined as textile materials made of parallel laid, cross laid or randomly laid webs bonded with the application of adhesive or thermoplastic fibers under the application of heat and pressure. In nonwoven production, the fibers are directly converted into a textile material. The important advantage of nonwovens is the higher production rate compared to conventional fabric formation such as weaving or knitting, since all yarn preparation steps are eliminated [31]. The nonwoven manufacturing process requires that the most efficient production method and the right binder be selected to give the desired properties. Therefore, various processing methods have been developed in order to obtain textile material with specific properties. The classification of nonwovens depends on the

selected parameter, e.g., method of production, technology of raw materials, properties. Web-formed nonwovens can be classified as wet, spun (spun-bonded and meltblown) and dry-laid (card: parallel and cross, random air-laid). Based on web bonding we can classify nonwovens as mechanically, thermally or chemically bonded. The great variety of types of nonwovens according to web bonding are shown in Table 1 [32].

**Table 1.** Classification based on web bonding.

| | |
|---|---|
| Mechanical bonding | Needle punch |
| | Spun laced |
| | Stitch bonded |
| Thermal bonding | Calendaring |
| | Through air bonding |
| | Sonic bonding |
| Chemical bonding | Impregnating |
| | Foam coating |
| | Spraying |
| | Print bonding |

On the other hand, the nonwovens can also be classified according to their structural properties. In Scheme 1, the classifications of the nonwovens are shown according to types of webs and their forming techniques [32]. A web is a thin layer of fibers [33].

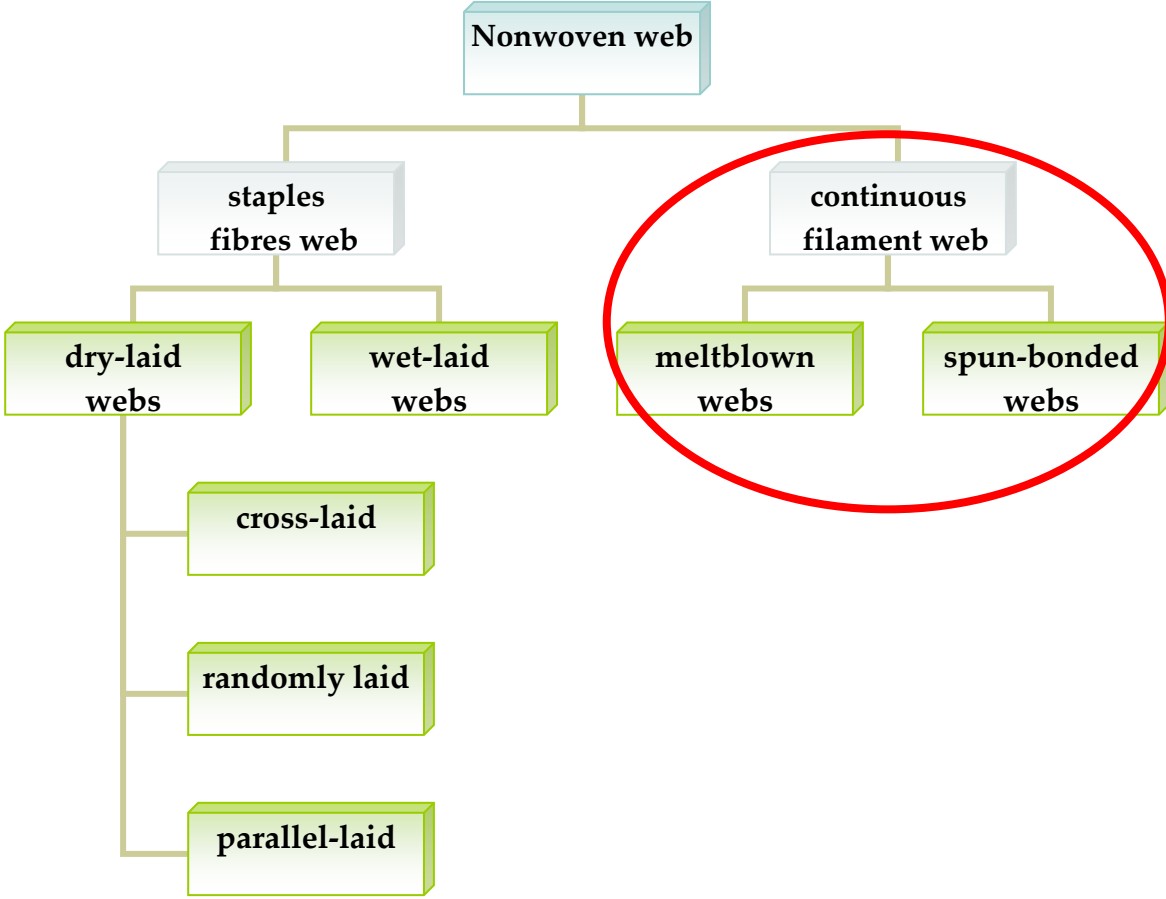

**Scheme 1.** Classification of nonwoven according types of webs.

Due to current ecological concerns, biodegradable polymers are processed into fibrous products, i.e., fibers, nonwovens, and fabrics. The paper will present the formation processes of nonwovens made from continuous filament web: spun-bonded and meltblown technologies. These technologies are good methods of processing thermoplastic biodegradable biopolymers such as polylactide, poly(hydroxyalkanoates), polycaprolactone and polybutylene succinate [34–40]. Spunbond and meltblown nonwoven webs technology uses the collection of continuous filaments produced by melt extrusion processes. The advantages of forming a continuous filament web are:

Flexibility to bond with other types of webs,

Resistance to shedding,

Relatively high mechanical strength such as higher tear strength,

Stability and resilience,

Lack of in textile character and feel,

Great possibility of modification e.g., antibacterial, antiflame properties [33,40].

Spun-bonded technology includes polymer melting, and then transportation and filtration of the polymer melt. The next steps are:

- Filament extrusion.
- Filament drawing.
- Filament deposition.

The process of making nonwovens is finished by bonding by using, e.g., calander (Scheme 2). The meltblown process involves the attenuation of the filaments using high-velocity hot air streams that impinge on the extruded filaments as they emerge from the extrusion nozzles to obtain much finer filaments. Finally, blown ultrafine fibres are collected on a conveyor belt with a vacuum underneath to generate a nonwoven web (Scheme 3) [40,41].

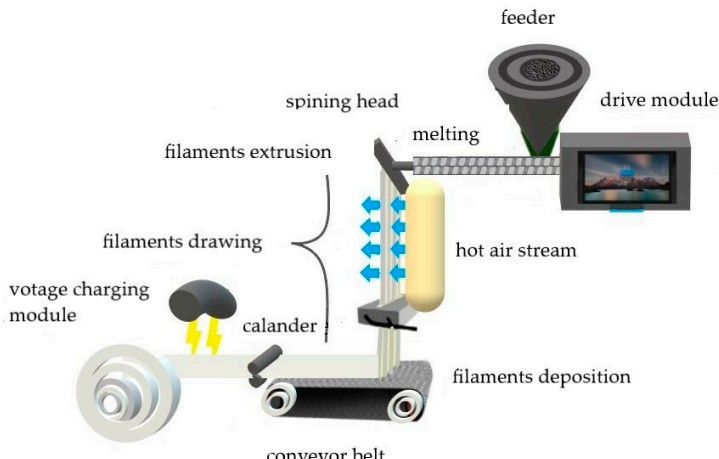

**Scheme 2.** Spun-bonded technology.

Depending on the techniques used, materials with different properties are obtained. The first and essential difference is the diameter of the fibers: in the meltblown technology, this is of the order of 1 μm, while the diameter of the spun-bonded fiber is approx. 20 μm (Figure 1). The meltblown technique uses much higher temperature and volumes of air, which allows the formation of microfibers. The meltblown web is softer, more delicate, and has smaller pore dimensions, which ensures better filtration efficiency. The spun-bonded web shows better mechanical performance because the fibers are thicker, continuous, and have random somewhat aligned orientation [34]. Meltblown nonwoven is used as filter material, due to the low fiber diameter, pore size, and areal density. Moreover, it is characterized by good thermal insulation and sorption capacity. Therefore, this material is used in the medical sector in respiratory protection masks, protective clothing, and filters. Meltblown material shows high filtration efficiency of bacteria, viruses,

solid particles, and harmful aerosols [42], whereas spun-bonded nonwoven has excellent physical properties, good tensile strength, elongation at break, and tear strength [33]. Therefore, both technologies are often used for the production of multi-layered nonwovens such as SMS (spun-bonded-meltblown-spun-bonded), SMMS (spun-bonded-meltblown-meltblown-spun-bonded). The hygienic or medical industry uses such compact materials for the production of disposable diapers, feminine care products, face masks, surgical scarfs, etc. [43–45].

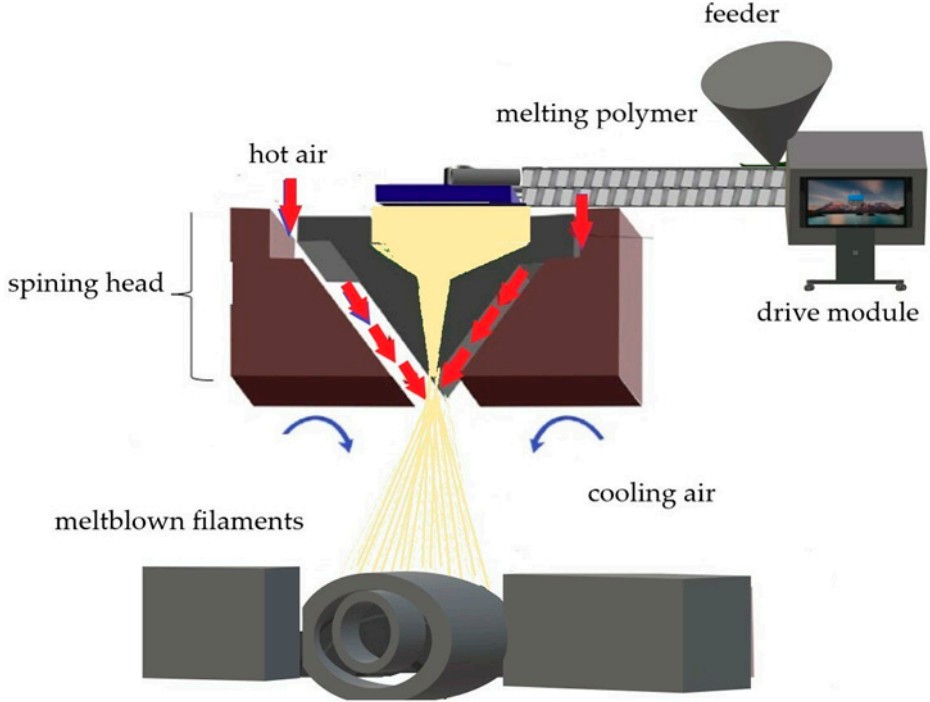

**Scheme 3.** Meltblown technology.

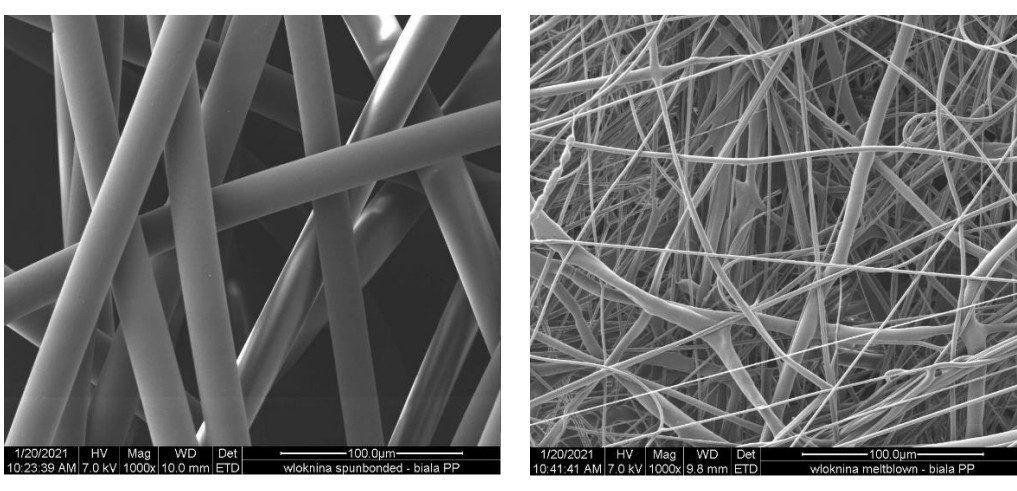

Spun-bonded nonwoven, 1000×       Meltblow nonwoven, 1000×

**Figure 1.** Difference in fiber diameter in spun-bond and meltblown technology.

Nonwoven production using melting technology allows obtaining a biodegradable polymer matrix for the application of active substances. Nonwovens made of biopolymers produced in the melting technology are a good textile base for the application of various types of active layers. Antibacterial substances are usually applied to the non-woven base, e.g., fosfonomycin [46], copper silicate [47], Ag nanoparticle [48], heterocyclic Nhalamine

acetate homopolymer [49], zinc [50]. Due to the SARS-CoV-2 pandemic, biodegradable non-wovens with antiviral properties are also currently under research [51–54]. Biodegradable nonwovens can be activated against pathogens by various techniques during the manufacture of the nonwoven, but also for the produced nonwoven. SEM images, obtained on the basis of own studies, on activated biodegradable nonwovens produced in the spun-bonded technique are presented in Figure 2.

Spun-bonded biodegradable nonwoven form aliphatic-aromatic copolyester with hyaluronic acid and plant extract

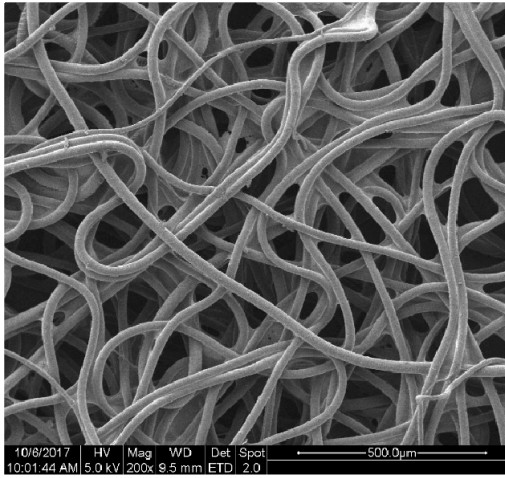

Spun-bonded biodegradable nonwoven form aliphatic-aromatic copolyester with protein

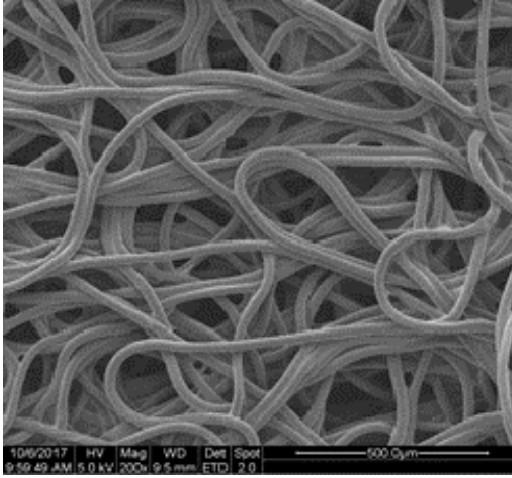

Spun-bonded biodegradable nonwoven form aliphatic-aromatic copolyester with polysaccharide polymer

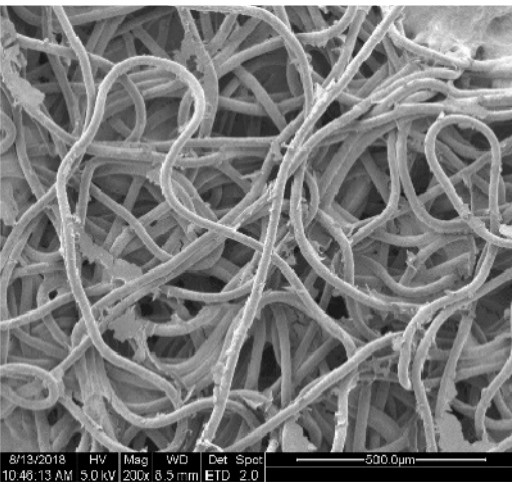

**Figure 2.** SEM images of spun-bonded biodegradable nonwoven after activation, 200×.

The nonwoven was activated by the padding method, which did not change the morphology of the fibers. The applied substances covered the fibers without damaging their structure [55]. The biodegradable spun-bonded nonwovens with active layers are shown in Figure 2.

### 3. Nonwoven Materials—Antipathogenic Modifications

Textile products, including nonwoven, can be modified by the use of active substances with an anti-pathogenic effect. The use of antimicrobial agents causes nonwovens to gain valuable properties, i.e., bacteriostatic, bactericide and fungicide. The interest in this area is due to the increased health awareness of pathogenic contamination. In the time of the SARS-CoV-2 pandemic, this is a very important aspect. There is a growing awareness of the need for protection against infections caused by pathogens and their adverse effects on personal hygiene and related health risks. Hence, in the last few years, research has been carried out to minimize the growth of microorganisms on textiles. Microbial infections are one of the leading causes of death worldwide, especially in primary healthcare facilities, where patients generally have a reduced immune resistance, which is associated with an increase in morbidity and a more severe course of various types of infections [56,57]. It is mainly caused by the presence of colonizing pathogenic microorganisms such as bacteria, viruses, and fungi, which may be present on the surfaces of medical devices, sanitary appliances, textiles, health products or water treatment systems [58]. In addition, most of the patients exposed to the microbes are already in poor health and unable to resist the further complications of the infection. The increase in drug resistance of pathogenic microorganisms is also a serious problem. The emergence of "multi-drug resistant" bacteria on the surfaces of medical devices and medical textiles increases the risk of repeated contamination, which can be harmful and dangerous to health, and even fatal [59,60].

The above problems with pathogens have led to numerous studies on the development of active layers. In order to inhibit infection with microorganisms, the surfaces of medical devices, including textiles, i.e., nonwovens, are covered with special antibacterial coatings, which protect patients against secondary bacterial infection and limit the growth of drug-resistant pathogens [61]. Depending on the antimicrobial agent used, as well as the type of fiber, its composition, surface structure, and texture, various surface modification methods (chemical and physical) are used to impart antimicrobial properties. The antimicrobial substance can be applied to the surface of the material at the stage of finishing synthetic and natural textile products using methodssuch as padding and spraying. Application can also be during the production of the material, e.g., as an additive to the base polymer during spinning. In the case of textiles, a good modification technique is surface grafting, the form of which strongly depends on whether the fiber is natural or synthetic, and also on its physicochemical properties.

Techniques for grafting textile surfaces include:

- chemical grafting;
- plasma induced vaccination using radiofrequency or microwave plasma;
- radiation induced vaccination that uses high energy radiation (e.g., $\gamma$-Co60 rays);
- light induced vaccination using an ultraviolet light source [62].

Depending on the method of applying the antimicrobial substance to the textile material, it can act in two ways, namely, contact and/or diffusion. In the case of a substance with a contact effect, it is placed on the fiber and does not disperse; it only works when the microorganism is in contact with the surface of the nonwoven material. In the case of diffusion, the active substance is on the surface of the material and migrates from the textile matrix to the external environment in order to deactivate the microorganisms (Scheme 4).

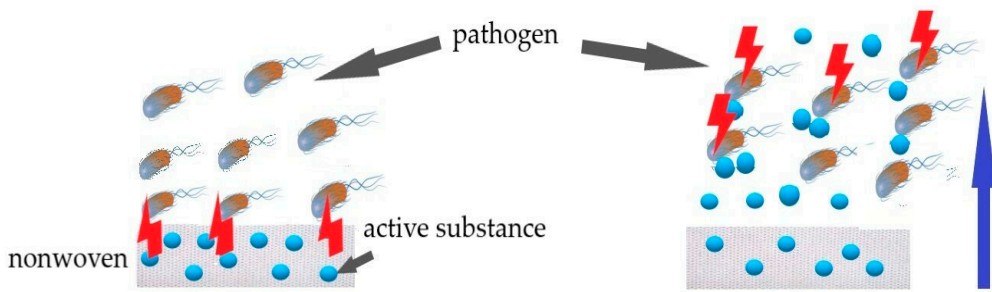

**Scheme 4.** Modes of action of active textile material on microorganisms.

Most of the antipathogenic agents used in commercial textiles are biocides and act differently according to their chemical structure and the level of affinity for specific target sites within a microbial cell.

As is shown in Scheme 5, the active substances can influence pathogens by:

- damage or inhibition of cell wall synthesis, which is crucial for the life and survival of the bacterial species;
- inhibition of cell membrane function, which can disregulate intra- and extracellular material flow and leak solutes important for cell survival;
- inhibition of protein synthesis, which is the basis of enzymes and cell structures, leading to the death of the organism or inhibition of its growth and multiplication;
- inhibition of the synthesis of nucleic acids (DNA and RNA) due to the binding of certain antimicrobial agents and inhibition of other metabolic processes, e.g., disruption of the folic acid pathway, which is necessary for bacteria to produce precursors important for DNA synthesis [63,64].

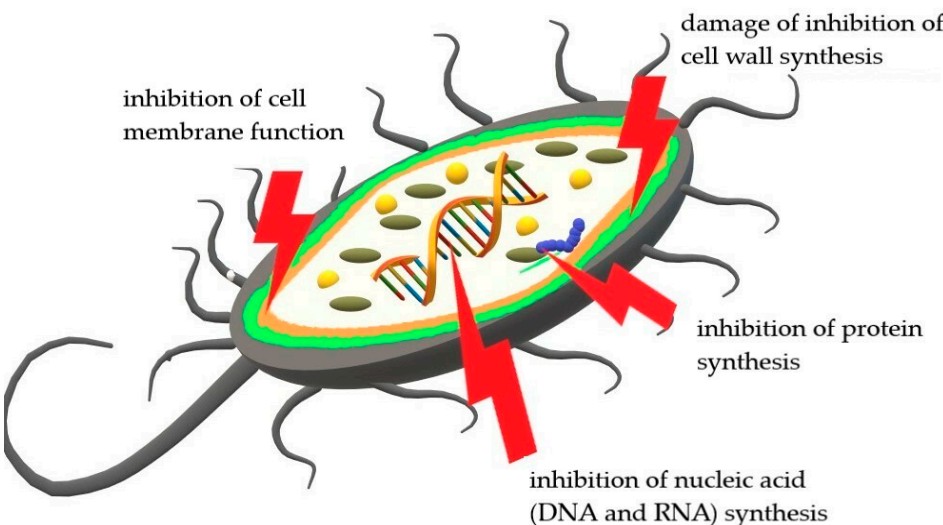

**Scheme 5.** The effect of active substances on the pathogen.

Active substances usually applied to the surface of textiles can be used, e.g., quaternary ammonium compounds, triclosan, and metal salts (such as copper, zinc, and silver) as such and in the form of natural polymers [65,66]. Quaternary ammonium compounds which have a positive charge on the N atom in solution are usually attached to the anionic surface of the fiber by ionic interaction. The antimicrobial effect is triggered by interactions between the cationic ammonium group and the negatively charged cell membrane of the microorganism. Quaternary ammonium compounds are removed from the structure of nonwovens over time due to the lack of strong bonding, resulting in a rapid decrease in its concentration. In the textile industry, another possibility, applied mainly for cotton,

polyester, nylon, and wool, is long-chain alkyl compounds (12–18 carbon atoms). These compounds are active against a wide range of microorganisms such as Gram-positive and Gram-negative bacteria, fungi, and some types of viruses [67].

Due to the antibiotic resistance of pathogenic bacteria, antimicrobial compounds extracted from herbs and plants have also been extensively studied as alternative therapeutic agents to combat microbial growth in nonwoven textiles. The main advantage of these compounds is effective antimicrobial action while ensuring safety, availability, non-toxicity to the skin and being environmentally friendly [68–70]. The effect of influent of plant extract on spun-bonded biodegradable nonwovens is shown in Table 2.

**Table 2.** Antipathogenic effect of plant extract and natural substance on spun-bonded biodegradable nonwovens.

| Samples | Microrganisms | Number of Mesophilic Microorganisms [cfu/g] | | |
|---|---|---|---|---|
| Time of exposure, days | | 0 | 3 | 28 |
| Biodegradable nonwoven with plant extract and polysaccharide | Bacteria | $3.3 \times 10^4$ | $4.9 \times 10^3$ | $<1.5 \times 10^1$ |
| | Fungi | $1.5 \times 10^1$ | $1.5 \times 10^1$ | $<1.5 \times 10^1$ |
| Biodegradable nonwoven with plant extract, polysaccharide and protein | Bacteria | $1.6 \times 10^4$ | $7.3 \times 10^3$ | $<1.5 \times 10^1$ |
| | Fungi | $1.9 \times 10^1$ | $1.5 \times 10^1$ | $<1.5 \times 10^1$ |

Activation of nonwovens can also be induced by the use of metals as antimicrobial agents. For natural fibers, this can performed as a final step. Another possible process is enzymatic surface modification of nonwovens, which increases the hydrophilicity and removes components from the surface, and also allows the introduction of a functional group to the surface of the material. The disadvantages of this technique are the slow diffusion of enzymes compared to common chemicals, limited temperature stability and slow speed reaction on synthetic materials. Some studies have shown that it is only by modifying the surface of the material's properties that the adhesion of bacteria to the surface can be reduced. Chemical antimicrobials can be used as a surface modifier. It is also necessary to carefully select the antimicrobial agent in terms of its temperature stability. Therefore, the most commonly used agents in this method are metallic particles, and even nanoparticles, because they do not degrade under the influence of standard processing conditions for the thermoplastic material of polymers [71].

Biopolymers from the group of polysaccharides are also used to modify the nonwovens. Chitosan is a natural and hydrophilic copolymer that is formed as a result of the deacetylation of chitin obtained from the exoskeleton of crustaceans such as crabs, krill, and shrimps, and the cell wall of some fungi and bacteria. It consists of two monomers: D-glucosamine and N-acetyl-D-glucosamine, connected by a β (1–4)-glycosidic bond. The antimicrobial activity of chitosan depends on its physicochemical parameters, i.e., average molar mass (Mw), degree of polymerization and of deacetylation but also pH of the medium and the type of microorganism. A pH value below the pKa of chitosan (<6.3) results in the formation of a polycation compound capable of interacting with negatively charged groups on the surface of microbial cells. Chitosan loses its antimicrobial properties at a pH above 7 due to the lack of protonated amino groups and low solubility. Low molecular mass chitosan can penetrate the cell wall, bind to DNA and inhibit mRNA synthesis, and thus protein synthesis. High molecular mass water-soluble chitosan has a higher density of positive charges, may cause leakage of some intracellular substances or may form an impermeable layer around the cell wall, blocking the transport of essential solutes into the cell [72–74]. The effect of chitosan on microorganisms is shown in Figure 3.

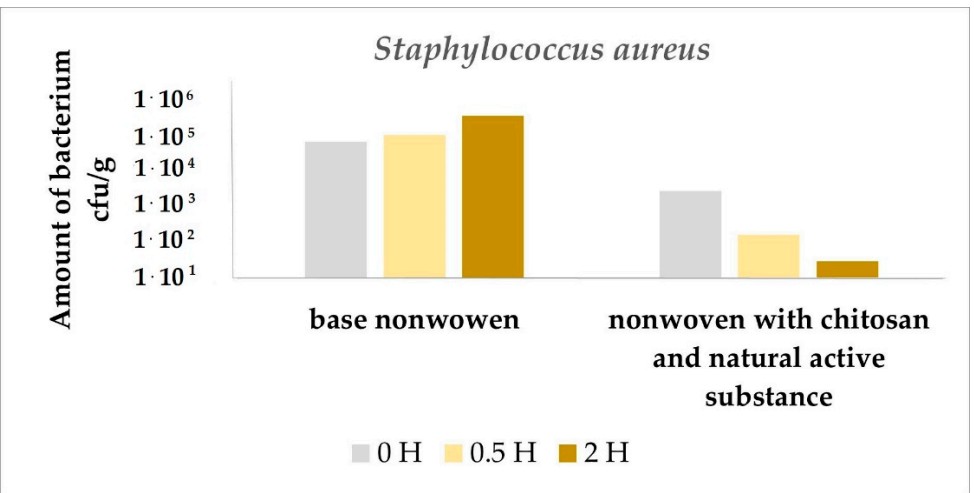

**Figure 3.** Effect of influence of chitosan on biodegradable nonwoven for microorganisms.

To obtain even more efficient and durable antimicrobial treatments for textiles, complexes based on chitosan and other more effective biocides have been developed [75]. Wang et al. created chitosan-metal complexes with divalent metal ions, including Cu (II), Zn (II) and Fe (II). The complexes showed broad antimicrobial activity against Gram-positive bacteria (*S. aureus* and *S. epidermidis*), two Gram-negative bacteria (*E. coli* and *P. aeruginosa*) and fungi (*Candida albicans* and *Candida parapsilosis*). The antimicrobial activity of the complexes was significantly higher than that of free chitosan or metal salts alone [76]. However, a major disadvantage of chitosan as an antimicrobial agent for finishing nonwovens is its degradability. During storage and temperature changes, some specific parameters of this polymer, such as viscosity and Mw, may change, which in turn influences the effectiveness of the antipathogenic activity.

## 4. The Biodegradation Process of Materials

Over the years, many definitions of biodegradation have been proposed. It was explained as 'an event that takes place through the action of enzymes or through chemical decomposition associated with living organisms [77]. Another definition presents the term as gradual breakdown mediated by specific biological activity [78]. Recently, it has been understood as the exothermic process of decomposition of organic substances by microorganisms into simpler substances such as carbon dioxide, water, and ammonia [79,80]. The reproduction of microorganisms depends on many factors, such as energy, the presence of oxygen, carbon, phosphorous, sulfur, nitrogen, calcium, magnesium, and other elements. The obtained energy is partially used by the microorganisms or lost as heat. The process can be observed especially in composting. Biodegradation can be conducted both under aerobic and anaerobic conditions [81]. A wide range of organisms participate in the breakdown of various materials, including bacteria, fungi, insects, worms, and many others.

Growing consumer awareness of ecology is driving the development of the market in this area. Many scientific terms are used in public spaces, often in the wrong way. It is extremely important to distinguish between biodegradation and degradation. Degradation is an irreversible process that affects the chemical structure. Abiotic degradation occurs under the influence of external factors (electromagnetic radiation, mechanical forces, etc.) [82]. Biodegradation refers mainly to biomass that can be actively consumed by microbial or enzymatic digestion, yet the similarity of some synthetic materials (e.g., green plastics) to natural ones is enough to enable their decomposition in this way [83]. The undeniable advantage of biodegradability of various materials is the potential reduction of waste. It should be highlighted that the process provides nutrients crucial for the growth of new life [83–85]. Nowadays, biodegradability is a recommended or required feature of everyday products and is essential for evaluating their sustainability [84].

*4.1. Selected Research Methods of the Biodegradation Process*

Numerous testing methods can be used to examine the biodegradability of various materials [86]. Some of them are presented in Table 3. The choice of method applied depends on the kind and properties of a sample. The biodegradation process can be conducted in three environments: water, soil, and compost. The latter is the most common matrix due to the way of handling waste. It should be pointed out that biodegradation does not occur for all materials by leaving them exposed to the elements. Therefore, industrial composting may be used to facilitate the breakdown of materials prone to the process in a controlled environment [83].

**Table 3.** List of selected biodegradation standards.

| Method | Title |
|---|---|
| ISO 11721-1:2002 | Textiles—Determination of resistance of cellulose-containing textiles to microorganisms—Soil burial test—Part 1: Assessment of rot-retardant finishing |
| ISO 11721-2:2005 | Textiles—Determination of the resistance of cellulose-containing textiles to micro-organisms—Soil burial test—Part 2: Identification of long-term resistance of a rot retardant finish |
| ISO 14851:2019 | Determination of the ultimate aerobic biodegradability of plastics materials in an aqueous medium—Method: measuring the oxygen demand by respirometer |
| ISO 14855–2:2007 | Determination of the ultimate aerobic biodegradability and disintegration of plastics under controlled composting conditions—Gravimetric measurement of carbon dioxide evolved in a laboratory-scale test |
| DIN EN 13432:2002 | Packaging—Requirements for packaging recoverable through composting and biodegradation—Test scheme and evaluation criteria for the final acceptance of packaging |
| DIN EN 14046:2003 | Packaging—Evaluation of the ultimate aerobic biodegradability and disintegration of packaging materials under controlled composting conditions—Method: analysis of released carbon dioxide |
| DIN EN 14047:2003 | Packaging of the ultimate aerobic biodegradability of packaging materials in an aqueous medium—Method: analysis of released carbon dioxide |
| DIN EN 14048:2003 | Packaging—Determination of the ultimate aerobic biodegradability of packaging materials in an aqueous medium—Method: measuring the oxygen demand in a closed respirometer |
| DIN EN 14995:2007 | Plastics—evaluation of compostability—test scheme and specifications |
| ASTM D5338-15 | Standard Test Method for Determining Aerobic Biodegradation of Plastic Materials Under Controlled Composting Conditions |
| ASTM D5929-18 | Standard Test Method for Determining Biodegradability of Materials Exposed to Municipal Solid Waste Composting Conditions by Compost Respirometry |
| ASTM D6006-17 | Standard Guide for Assessing Biodegradability of Hydraulic Fluids |
| ASTM D6400 | Standard Specification for Compostable Plastics |
| BS EN ISO 14851:2019 | Determination of the ultimate aerobic biodegradability of plastic materials in an aqueous medium. Method: measuring the oxygen demand in a closed respirometer |
| BS ISO 14852:2021 | Determination of the ultimate aerobic biodegradability of plastic materials in an aqueous medium. Method: analysis of released carbon dioxide |

Biodegradability of materials has become a desired feature due to the growing problem connected with waste management. Land filling, recycling, and incineration are among the conventional methods for fabric waste management. A green alternative to the listed methods can be large scale composting, which is characterized by the presence of significant microbial and enzymatic active components as well as controlled temperature (50–65 °C) [87,88]. Biodegradability and compostability is crucial due to the growing problem with microplastics, which are present in rivers, oceans, lakes, and sediments [89,90].

The fabric and nonwoven industry is very diverse. Fully natural or synthetic materials, as well as mixtures of these, are available. The composition has a great impact on the biodegradability/compostability of the product. Li et al. [87] aimed to compare the biodegradation process of cotton jersey fabrics with three levels of finishing treatment, such as scoured and bleached, softener added, and resin added. Additionally, the polyester jersey fabric was examined. The authors performed studies based on the measurement of $CO_2$ release and weight loss in a soil environment. Simultaneously, a compostability test was performed in a composting facility. The conducted study revealed different biodegradability of the tested materials. Polyester fabric showed a slight initial degradation, yet the fabric stayed intact after testing. The level of finishing treatment also influenced the breakdown. The presence of softener accelerated the degradation rate, while fabric with resin showed a slow degradation rate. Cotton samples were confirmed to be compostable. A similar study [91] showed the impact of common finishes on the biodegradability of cotton fabrics. Results were evaluated on the basis of the production of $CO_2$ in a soil environment. The obtained results showed that untreated fabrics and non-crosslinked finished materials were more prone to biodegradation (40–60% weight loss) than crosslinked samples (<20% weight loss) over the same period of time.

Shedding microfibers during laundering prompted scientists to conduct studies of biodegradation in the water environment [92]. Tests revealed that microcrystalline cellulose is most susceptible to breakdown, followed by cotton, rayon, polyester/cotton, and polyester. In a matrix composed of activated sludge at low concentration and lake water, cotton and rayon yarns showed 70% biodegradation, whereas in sea water about 50% biodegradation was reached. For polyester, degradation was not noticed.

Other research [93] was focused on the influence of fabric structure on biodegradation processes. A wide range of samples was tested, including different combinations of yarn count, weave structure, weave density, presence of dye, water-repellent, and peach skin. Authors concluded that the higher yarn count or wave density the lower the release of fibers. The fabric with peach skin released more fibers. The presence of water-repellent finish released fewer fibers with shorter length. The obtained results showed the influence of the structure on the emission of cotton fiber into the environment. Additionally, the study revealed that the degradation of cotton fibers is lower in seawater than in soil and depends on the temperature and structure. The untreated fabric was more prone to biodegradation.

The low polyester biodegradability may result from the fact that typical microbial communities in the environment do not attach to polyester [94]. Additionally, the polymer is less prone to disintegration by hydrolysis due to its low moisture regain and high hydrophobicity [92,95].

Cellulose-based yarns are relatively prone to biodegradation in various environments. Rayon fibers have lower crystallinity and orientation, and higher moisture regain which result in higher break down than cotton fibers [92]. Microcrystalline cellulose is characterized by high crystallinity, yet it is prone to biodegradation. This may arise from its small particle size.

The SARS-CoV-2 pandemic forced the use of masks, which directly led to the generation of a large amount of waste. The most commonly used masks were based on polypropylene and were a hazard for soil and aquatic ecosystems. The progressive awareness within society of issues in the field of ecology forced scientists to develop new base

compositions that would be more environmentally friendly. Deng et al. [96] described biodegradable, antiviral, and antibacterial cellulose nonwovens as a good solution in light of the increasing pollution of the environment. The authors confirmed that usage of low doses of antiviral/antibacterial components will not influence biodegradability of cellulose materials.

Recently, the application of PLA has aroused interest due to its biodegradability. It is prone to break down and can be compared to plasticized starch. However, the biodegradation of PLA requires a higher temperature than starch, which prevents home composting which is characterized by lower temperatures [97]. The degradation of plastics depends on many factors, including properties of the polymer, chemical structure, molar mass, glass transition temperature, melting point, polydispersion, crystallinity, sample surface, etc. [98–100]. A study on the biodegradation of PLA showed that the higher the degree of crystallinity, the molar mass and the melting point, the slower is its degradation. While in the first stage hydrolytic degradation can be observed, the second stage is related to the activity of microorganisms activity [98]. Liu et al. [101] investigated biodegradation of PLA/PHB-blended nonwovens in the presence of microbial communities and indicated the role of Proteobacteria and Firmicutes in the process in a soil environment.

As noted above, the biodegradability of products is a desired feature. A wide range of legal documents deals with the biodegradability/compostability of different types of products: packaging, plastics, fibers, etc. The focus is on plastics, which are the main pollutant of the marine and land ecosystems [82]. According to the literature [102], the mass of plastic has reached 8 billion tonnes globally in 2020, which is twice the living biomass. The necessity for better plastic waste management in order to avoid pollution resulted in the formulation of legal and policy documents, such as a list of Sustainable Development Goals [103], a Green Paper 'On a European Strategy on Plastic Waste in the Environment' [104], the European Strategy for Plastics in Circular Economy [105] and Directive on 'The reduction of the impact of certain plastic products on the environment' [106]. Most of the documents indicated the main challenges in proper waste management and recycling. The necessity of further explanation of the term 'biodegradable' was highlighted.

### 4.2. Biodegradation of Nonwovens in Compost Environment—Laboratory Scale

The biodegradability (microbiological decomposition) of nonwovens (100% viscose, cotton, bamboo, PLA) was tested in a compost environment in accordance with an accredited procedure based on international standards: PN-EN 14045:2012, PN-EN 14806:2020, PN-EN ISO 20200:2016-1, PN-EN 14995:2009, PN-EN 13432:2002. The biodegradation studies were carried out under aerobic conditions in the compost environment under controlled temperature and humidity conditions ($58 \pm 2$ °C; 80%) simulating the natural processes occurring during decomposition. The humidity control was performed daily. The medium used was obtained from an industrial composting plant. Before starting the tests, the microbiological activity of the test inoculum and its humidity were determined ($\geq$106 cfu/mL; 40–65%). The biodegradation process was monitored by removing individual samples from the inoculum at specified intervals, washing, and drying to a constant weight. The relative weight loss was then determined.

The study showed varied biodegradability for nonwovens made of viscose, cotton, bamboo, and PLA. Materials that were 100% viscose and bamboo showed the greatest susceptibility to microbial decomposition (100% biodegradation after 8 weeks). Cotton nonwovens reached 100% biodegradation after 24 weeks in the compost and the PLA sample reached a maximum of 73%. The obtained results are presented in Figure 4.

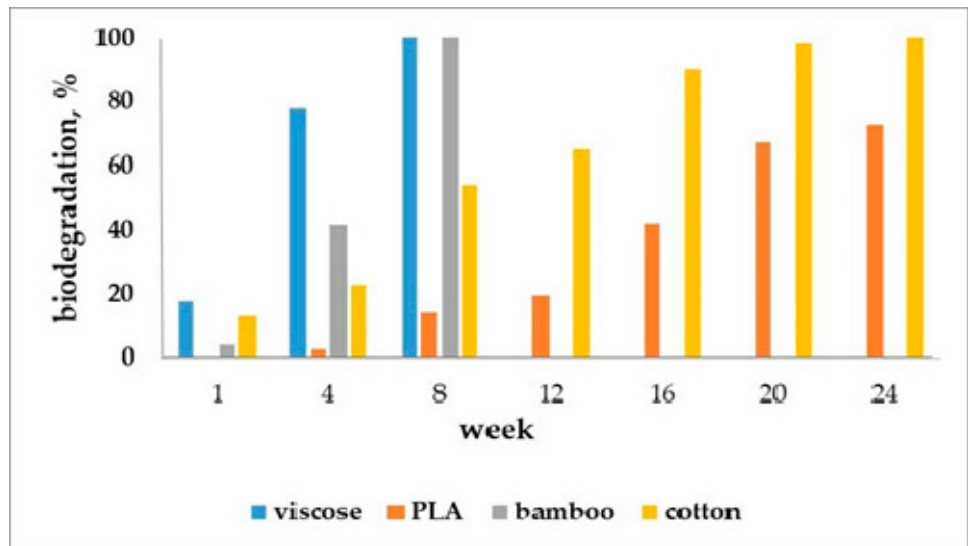

**Figure 4.** Comparison of biodegradation progress of tested nonwovens.

Many factors influence the biodegradation process, including crystallinity, construction of a polymer chain, presence of additives, surface area to volume ratio, etc. [82]. The composition, weave density and finish also influence microbial degradation and bacterial adherence on different textile fabrics. The study conducted by Bajpai et al. [94] revealed diverse adherence of bacteria on fabrics made of cotton, polyester filaments, and polyester (staple)-cotton blended yarn. The maximum adherence was found in cotton, followed by the adherence blend. The scientists pointed out that surface morphology also plays a crucial role during adherence. It can be concluded that the lower adherence is associated with lower biodegradability. Biodegradable nonwovens additionally modified with active substances may show a completely different rate of degradation. The addition of substances of plant origin enhances the degradation process by microorganisms in compost, as shown in Figure 5. The substances used were environmentally friendly and made the material more accessible to the compost microflora. The base of the nonwovens was hydrophilic, and the addition of the substance caused its water-binding capacity to increase, which contributed to its degradation.

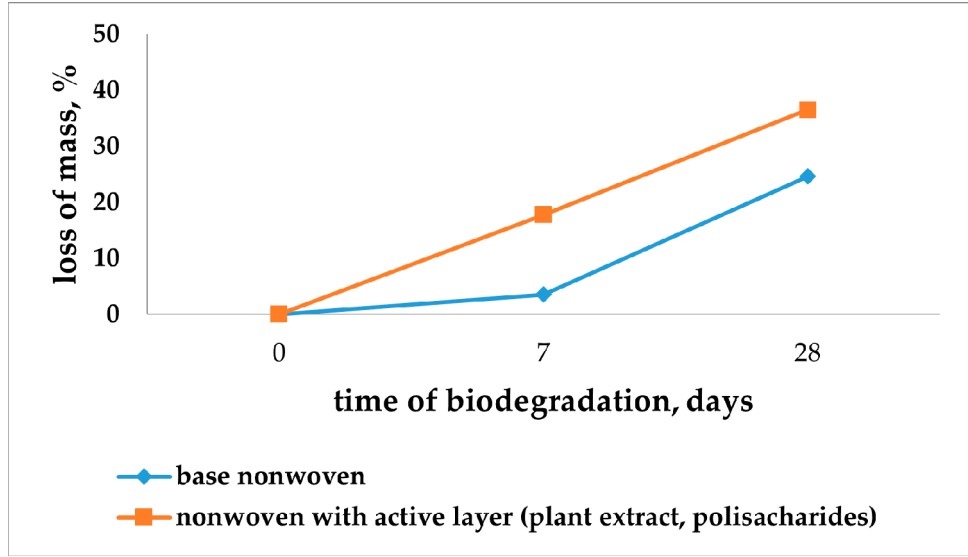

**Figure 5.** Biodegradation in a compost biodegradable spun-bonded nonwoven with active layer.

## 5. Antipathogenic Activity—Activity Evaluation

Biodegradable nonwovens are almost indistinguishable from their traditional counterparts made from petroleum raw materials and therefore their antimicrobial activity should be assessed using the same methods. When selecting the test, one should take into account the properties of nonwovens such as hydrophilicity, the ability to diffuse the active ingredient into the substrate, and its concentration. This information is needed to determine whether the tested nonwoven fabric should be tested by a qualitative or quantitative method. In this part of the article, we will try to briefly introduce the differences between these test methods.

### 5.1. Qualitative Methods

Qualitative tests allow assessing the antimicrobial activity of the nonwoven fabric if the antipathogenic component diffuses into the substrate to a minimum degree. This means that if the active layer on the sample is not soluble in water, the tested object will not show antimicrobial properties. The results for the qualitative methods are given in a descriptive form; only with a high concentration of the active substance and the formation of a zone of inhibition of growth around the sample can the result be obtained in the form of a numerical value. They are most often used for screening tests, allowing only those with the highest antimicrobial activity to be selected from a large number of samples. The qualitative methods are not only fast but also easy to implement. They can be used to test a large number of samples. Table 4 shows the most popular qualitative methods of assessing antimicrobial activity.

**Table 4.** Qualitative methods for the assessment of antimicrobial activity.

| Numer Normy | Name of the Standard | Test Organisms |
|---|---|---|
| AATCC 147 [107] | Antibacterial Activity Assessment of Textile Materials: Parallel Streak Method | *Stapyloccocus aureus* *Klebsiella pneumoniae* |
| PN-EN ISO 20645 [108] | Textile fabrics-Determination of antibacterial activity-Agar diffusion plate test | *Stapyloccocus aureus* *Klebsiella pneumoniae* *Escherichia coli* |
| PN-EN 14119 [109] | Textile fabrics-Determination of antibacterial activity-Agar diffusion plate test | *Aspergillus niger* *Chaetomium globosum* *Penicillium pinophilum* *Trichoderma virens* *Paecilomyces variotii* |
| ATCC 30 [110] | Antifungal Activity, Assessment on Textile Materials: Mildew and Rot Resistance of Textile Materials | *Aspergilus niger* *Penicillium varians* *Trichoderma viride* |

Qualitative tests can be used to test not only bacteria but also fungi. Mold fungi not only cause disease in humans but also break down textiles. Some qualitative tests permit assessment of the resistance of the fabric to biodeterioration. Quantitative methods do not permit this.

In this article we will present two of the interchangeable quality methods: AATCC 147 [107] and PN-EN ISO 20645 [109]. In these standards, the test organisms are standard strains of Gram (+) *S. aureus* and Gram (−) K. pneumoniae and *E. coli* (only PN-EN ISO 20645). *Staphyloccocus aureus* is a bacterium that causes wound infections and nasopharyngeal infections, whereas Klebsiella pneumonia causes pneumonia, *and Escherichia coli* can cause diseases of the digestive and urinary systems. However, both standards allow testing against other test organisms. Figure 6 presents exemplary results of testing one biodegradable nonwoven fabric soaked with Penicillin G 0.03 units, made accor×ding to two standards AATCC 147 (a) and PN-EN ISO 20645 (b).

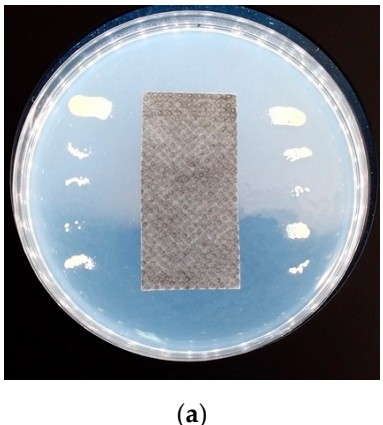
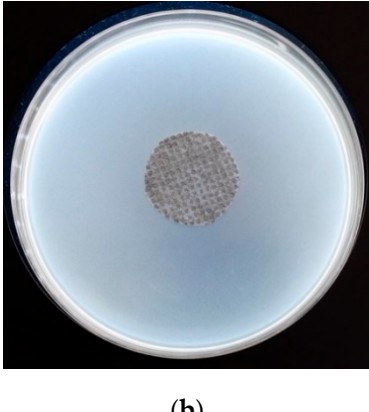

| (**a**) | (**b**) |

**Figure 6.** Qualitative methods of assessing antibacterial activity: (**a**) AATCC 147 against *S. aureus* and (**b**) PN-EN ISO 20645 against *E. coli*.

Figure 6a shows the test against *S. aureus*, which was linearly spread over TSA agar. In the AATCC 147 [107] method, bacteria are initially incubated in broth for 24 h at 37 °C, and then a suspension is prepared by dilution with sterile distilled water. Using a 4 mm inoculum loop, one loop of the diluted inoculum is loaded and transferred to the surface of a sterile agar plate, making five strips covering the center area of a standard petri dish without refilling the loop. The lines are made by dipping the loop in suspension once, which results in a reduced number of bacteria on each successive line. Test samples with dimensions of 25 × 50 mm are placed perpendicular to the line on the prepared plates. The entire surface of the fabric must be flush with the agar surface. Plates are incubated for 18–24 h at 37 °C. After this time, the growth under the sample and the presence or absence of a zone of growth inhibition are assessed. The method requires comparing the growth of bacteria under the test sample to samples with known bacteriostatic activity and to the same untreated samples. The standard provides only two results: antibacterial activity if there are no bacterial colonies directly under the sample in the contact area, and no antibacterial activity if there are bacterial colonies. The method does not predict activity staging, although the size of the zone of inhibition and the narrowing streaks allow the estimation of residual antimicrobial activity. The standard recommends that evaluation criteria be agreed upon by the interested parties before the test is performed. In Figure 6a, the zone of growth inhibition is clearly visible, which proves the activity of the tested nonwoven fabric against the *S. aureus* pathogen.

Figure 6b shows the result of testing the antibacterial activity of the sample against *E. coli*, performed in accordance with the guidelines of PN-EN ISO 20645 [108]. The bacteria are picked up on TSB broth and incubated for 16–24 h at 37 °C. The cultures are passaged 3–4 times "liquid to liquid" maintaining the same incubation conditions. This suspension is diluted so as to obtain a density of $1–5 \times 10^8$ cfu/mL. 10 mL of TSA agar at 45 °C is poured onto petri dishes and allowed to solidify. Then 1 mL of the previously prepared bacterial suspension is added to 150 mL of TSA agar at 45 °C. After mixing, the agar is dispensed 5 mL each into plates containing 10 mL TSA solidified medium. The textile sample with an average of 25 mm is placed so that its entire surface touches the agar surface. The prepared plates are incubated for 18–24 h at 37 °C, the same as for the method described in the AATCC 147 standard. After this time, an assessment of the growth of the microorganisms on the agar under the test sample and around it is performed. The PN-EN ISO 20645 standard distinguishes 3 levels of activity assessment depending on the growth of bacteria under the test sample compared to the control growth and the size of the zone of growth inhibition. We distinguish the following grades of assessment: insufficient efficacy, limited efficacy, and good effect. There is no visible zone of growth inhibition in Figure 6b in the photo. The sample, however, shows a good antibacterial effect against *S. aureus* because when it is removed from the agar, no bacterial growth is visible under the sample.

The experiment presented in the paper illustrates the differences between two qualitative tests: PN-EN ISO 20645 and AATCC 147. The result of testing the same sample, biodegradable nonwoven fabric, soaked with the same killer—Penicillin G antibiotic, is the same for both methods, i.e., the tested sample shows antipathogenic activity. Visually, you can see large differences in the growth of bacteria around the sample in both methods. In the AATCC 147 method, a wide zone of growth inhibition is visible around the sample, while in the method described in the PN-EN ISO 20645 standard, there is no such zone, and bacterial growth is inhibited only under the sample. These differences result from the method of inoculating the agar with bacteria which are at surface level in the AATCC 147 standard and deep in the PN-EN ISO 20645 standard, and from the type of strains used for each method. The microbial density is also different. These features influence the result of the test of antibacterial activity, which, at a lower concentration of the active agent in the sample, may give a different result depending on the method used. When comparing the results of qualitative tests for antimicrobial activity, special attention should be paid to the test method and test strains and evaluation should only be made against the same standard.

*5.2. Quantitative Methods*

Quantitative methods are methods in which the number of bacteria on the samples is determined, and the result of the sample activity is presented in the form of numerical values, e.g., logarithms and percentages. The tests performed with these methods, for the most part, consist of applying the suspension of bacteria of the standard strains to the test sample, incubation and, after dilution, determining the number of bacteria by the plate method. In these methods, a very important property is the hydrophilicity of the sample, because the contact surface of microorganisms with the sample is of major importance for the test results. Table 5 presents the most popular quantitative research methods.

**Table 5.** Quantitative methods for assessing antimicrobial activity.

| Number of the Standard | Name of the Standard | Test Microorganisms |
|---|---|---|
| PN-EN ISO 20743 [111] | Textiles-Determination of antibacterial activity of textile products | *Staphylococcus aureus* *Klebsiella pneumoniae* |
| ASTM E2149 [112] | Standard Test Method for Determining the Antimicrobial Activity of Antimicrobial Agents Under Dynamic Contact Conditions | *Escherichia coli* |
| JIS L 1902 [113] | Testing for antibacterial activity and efficacy on textile products | *Stapyloccocus aureus* *Klebsiella pneumoniae* *Escherichia coli* *Pseudomonas* |
| AATCC 100 [114] | Test Method for Antibacterial Finishes on Textile Materials: Assess | *Stapyloccocus aureus* *Klebsiella pneumoniae* |
| ASTM E 2180 [115] | Standard Test Method for Determining the Activity of Incorporated Antimicrobial Agent(s) in Polymeric or Hydrophobic Materials | *Stapyloccocus aureus* *Klebsiella pneumoniae* *Pseudomonas aeruginosa* |

The adsorption method described in the PN-EN ISO 20743 standard is one of the most popular research methods used to assess the antibacterial activity of hydrophilic textiles [111]. It consists of inoculating a sample with a mass of 0.4 g of the tested sample with a suspension of bacteria with a density of $1–3 \times 10^5$ cfu/mL and a volume of 0.2 mL, and incubation under static conditions for 18–24 h at 37 °C. The inoculum is prepared in NB broth 20 times diluted in distilled water. Samples, in triplicate, are washed, diluted, and quantified by the plate method on a solid nutrient medium immediately after inoculation, and another three times after incubation. The test result is presented in the form of a logarithmic value of the antibacterial activity. This is the difference between the growth of bacteria on the control samples measured at 0 h and 24 h and the growth of bacteria

on samples containing antipathogenic agents. Figure 7 shows a diagram of the method's execution. The PN-EN ISO 20743 method allows for detection of even a slight antibacterial activity and is sensitive to slight changes in the concentration of the active agent.

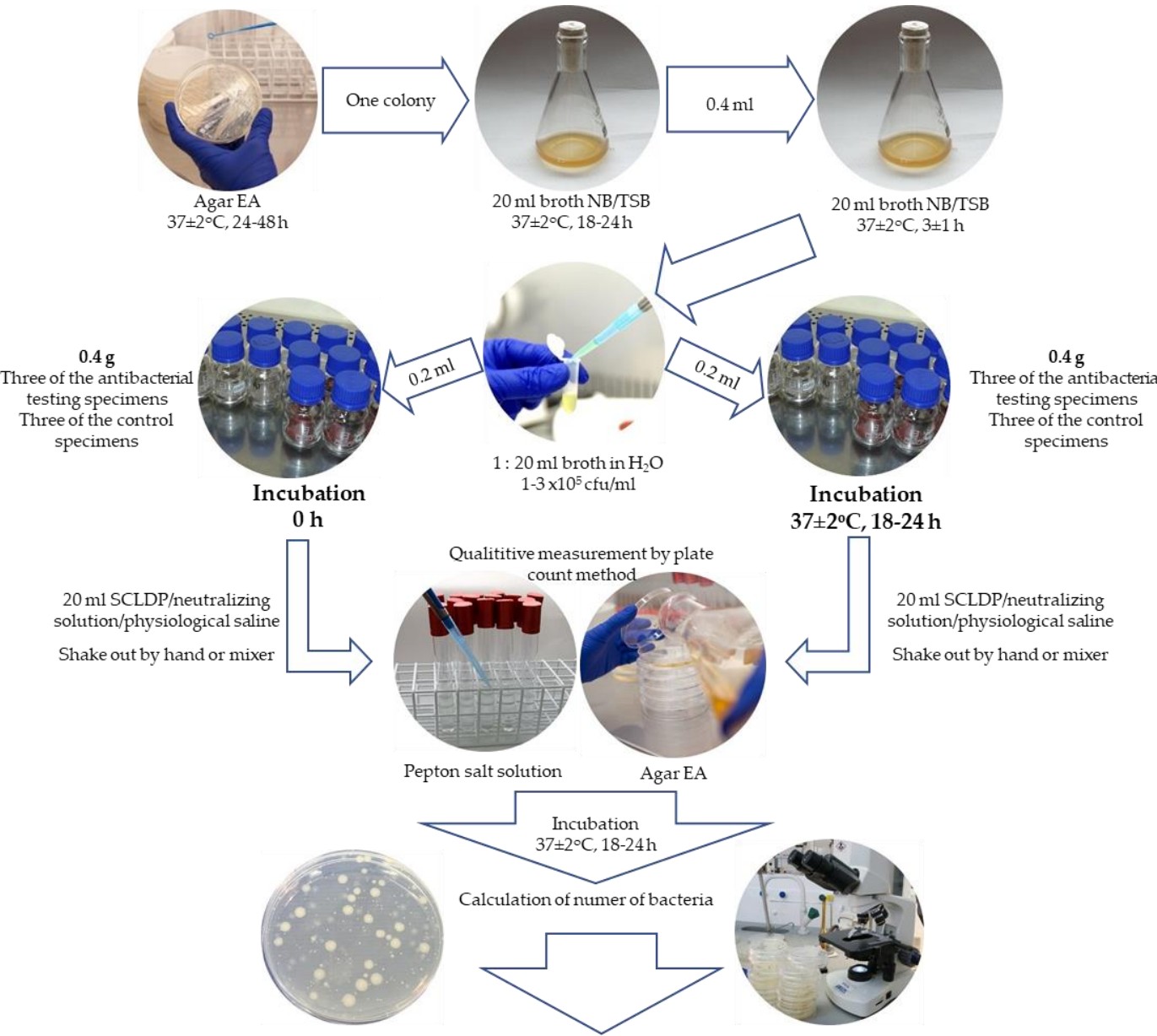

**Figure 7.** Scheme of the test according to the PN-EN ISO 20743 absorption method.

The method described in the JIS L 1902 [113] standard is so similar to the PN-EN ISO 20743 method that we will not discuss it in detail in this article. The sample weight, amount and density of the bacterial suspension as well as the medium and incubation conditions in both standards are identical.

We will briefly present the ASTM E2149 standard, in which the tested sample with a mass of 1 g is incubated under dynamic culture conditions in a phosphate buffer with a bacterial density of $1.5^{-3} \times 105$ cfu/mL at a temperature of 35 °C for 1 h [113]. The density of the bacterial suspension is determined by the plate method after the contact of the microorganisms with the test sample and the control sample. On this basis, a value for the logarithmic and/or percentage reduction is calculated. Each sample is tested in one repetition and is plated out of each dilution into three petri dishes. Use of alternate

organisms will be included in the report, in addition to any other modification of media, buffer, bacterial concentration, etc. This dynamic shake flask test was developed for routine quality control and screening tests in order to overcome difficulties in using classical antimicrobial test methods to evaluate substrate-bound antimicrobials.

In the literature, one can find comparisons of methods for assessing antibacterial activity and differences resulting from the specificity of their implementation and the impact on the test results [116]. As presented by Kaźmierczak and al., the activities of PLA nonwovens were assessed by two quantitative methods: JIS L 1902 and ASTM E2149 [117]. The nonwoven fabric showed the highest activity when assessed according to the JIS L 1902 method and showed no percentage reduction when assessed according to the ASTM E2149 method. Therefore, when choosing a research method or comparing test results, particular attention should be paid to the research method.

Selection of appropriate test methods can be of great importance in assessing antimicrobial activity. Obtaining the result of antibacterial activity in quantitative methods does not mean obtaining a similar result in qualitative methods. On the other hand, the assessment of bactericidal activity obtained with qualitative methods, in almost all cases, will show a similar result with quantitative methods. When comparing the test results, particular attention should be paid to the test method; therefore, knowledge of the test methodology enables the selection of the most suitable standard for a given product.

## 6. Summary

Environmental protection is a necessary measure for human development. The growing problem with the deposition of waste has made biodegradable nonwoven materials more and more often an alternative to the commonly used, difficult to recycle plastic in use until recently. The application of biodegradable materials is undoubtedly a very important element of environmental protection. Biodegradable nonwoven materials are used in many sectors, but the dominant one is the medical industry, mainly associated with the production of specialized materials. Antimicrobial layers on nonwovens give them special properties, such as bacteriostatic, bactericidal, or fungicidal properties. Giving textile products antimicrobial properties has been used in medical devices and in the production of, among others disposable masks, wound dressings, or gowns. Depending on humidity, nutrients and temperature, microorganisms can contaminate textiles. In order to prevent contamination with microorganisms, the surfaces of nonwoven materials are covered with special antibacterial coatings. On the one hand, these protect patients against secondary bacterial infection. On the other hand, they limit the growth of drug-resistant pathogens by reducing the use of drugs. They also have a beneficial effect on reducing environmental contamination with antibiotics. The antipathogenic activity can be determined using many research methods.

**Author Contributions:** Conceptualization, participation in literature review, writing—original draft preparation of the manuscript and editing, L.M.-K.; participation in literature review, writing—original draft, and preparation of the manuscript, K.G.-J.; participation in literature review, writing—original draft, and preparation of the manuscript, J.J.-P.; participation in literature review, writing—original draft, and preparation of the manuscript, M.D.; writing—original draft and preparation of the manuscript, M.W.-W. All authors have read and agreed to the published version of the manuscript.

**Funding:** This research received no external funding.

**Institutional Review Board Statement:** Not applicable.

**Informed Consent Statement:** Not applicable.

**Data Availability Statement:** Not applicable.

**Conflicts of Interest:** The authors declare no conflict of interest.

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
