# Peer review of "Biodegradable Nonwoven Materials with Antipathogenic Layer"

_environments, doi:10.3390/environments9070079_

Round 1
Reviewer 1 Report
The review is content rich and will be of interest to researchers in the related field. However, the presentation of the content needs extensive improvement. Specially the language, grammar and figures. Below are some examples, and they are far from an exhaustive list. For publication, the authors are strongly recommended to seek advice to improve the presentation.
- Line 34: A typo: "polilactide"
- Introduction should be broken down into more paragraphs. Hard to read. The sentence-to-sentence transition needs to be improved. At its current stage, it is difficult to follow the line of thoughts.
- Scheme. Typo.
- Fig 3. Typo in the x-axis. What is “jtk” on the y-axis? Y-axis line is missing.
- Fig 1. Citation is missing.
- Line 218: An example of poor placement of punctuation marks.
- Line 228: it should be “the active substances can influence by”
- Line 250-251.
- Line 270-273. Lack of citation. The sentence construct is problematic.
- What is the basis of the selected methods in Table 2?
- The graphs format needs consistency throughout the manuscript. Typography, font, line color, line thickness, etc.
Author Response
Response to reviewer 1
Replies to comments and suggestions
English language and style
Extensive editing of English language and style required
English language was edited for proper grammar and overall style.
Comments:
- Line 34: A typo: "polilactide"
"polilactide" changed to polylactic
- Introduction should be broken down into more paragraphs. Hard to read. The sentence-to-sentence transition needs to be improved. At its current stage, it is difficult to follow the line of thoughts.
The introduction has been divided into more paragraphs as suggested by the reviewer.
- Scheme. Typo.
Removed a typo in the schematic.
- Fig 3. Typo in the x-axis. What is “jtk” on the y-axis? Y-axis line is missing.
Fig. 3. The entry on the X axis has been corrected. Changed the notation "jtk" to cfu on the y axis? Y axis line added.
- Fig 1. Citation is missing.
The presented materials come from own research, therefore no citation was provided.
- Line 218: An example of poor placement of punctuation marks.
- Line 228: it should be “the active substances can influence by”
- Line 250-251.
Changes were made as suggested by the reviewer
- Line 270-273. Lack of citation. The sentence construct is problematic. All capital had the same literature nr. 72.
The entry was changed to:
Some studies have shown that only by modifying the surface of the material's properties, it is possible to reduce the adhesion of bacteria to the surface. Chemical antimicrobials can be used as a surface modifier.
- What is the basis of the selected methods in Table 2?
Table 2 presents selected examples of standards that are used to test the biodegradability process. The authors' aim was to present how wide the scope and applicability of the standards is with regard to various materials.
- The graphs format needs consistency throughout the manuscript. Typography, font, line color, line thickness, etc.
The format of the graphs has been standardized throughout manuscript

Reviewer 2 Report
I think this is a correct review articleAuthor Response
Thank you for positively assessing the article.
Reviewer 3 Report
The manuscript entitled Biodegradable nonwoven materials with antipathogenic layer emphasizes the importance of biodegradable textile materials, especially nonwovens materials prepared by melt-blown or spun-bond technologies, with an antipathogenic layer.
The pandemic undoubtedly showed the importance of such a review.
The manuscript needs English editing. There are many difficult-to-understand passages.
Section 2. Biodegradable nonwovens are mainly about the technologies, properties of nonwovens, applications, and their advantages. Due to the section is not about biodegradability, I suggest titling the section more adequately.
Scheme 1 seems to be incomplete. Are not there other types of webs composed of continuous filaments? I understand that the manuscript is about the SB and MB nonwovens. However, such a scheme should be general, so I suggest mentioning other techniques for the production of nonwovens.
Are the figures in this manuscript from the archive of the authors? Or are they taken from other articles? Please indicate this under the articles and provide a reference if appropriate.
In subsection 4.2, the study shows various biodegradability of nonwovens has no citation. Is this study of the authors? Please, clarify whether it is an original article or a review; many passages look to be original articles or just the opinions of the authors?
Please, use the internationally used units, replace jtk with CFU.
There is no control missing; however, extensive text editing is needed.
Author Response
Response to reviewer 3
Replies to comments and suggestions
English language and style
Moderate English changes required
English language was edited for proper grammar and overall style.
Comments:
- The manuscript needs English editing. There are many difficult-to-understand passages.
English language was edited for proper grammar and overall style.
- Section 2. Biodegradable nonwovens are mainly about the technologies, properties of nonwovens, applications, and their advantages. Due to the section is not about biodegradability, I suggest titling the section more adequately.
The title of chapter two has been changed to Technologies of producing the biodegradable nonwovens.
- Scheme 1 seems to be incomplete. Are not there other types of webs composed of continuous filaments? I understand that the manuscript is about the SB and MB nonwovens. However, such a scheme should be general, so I suggest mentioning other techniques for the production of nonwovens.
In the second chapter of the article, information on other techniques for producing nonwovens was added. Table 1 was also added.
The classification of nonwoven depending on the selected parameter e.g.: method of production, technology of raw materials, properties, may be different. Based on production techniques: web formation nonwovens can be classification of wet, spun (spun-bonded and meltblown) and dry –laid (card: parallel and cross, random air-laid). Based on web bonding we can classify nonwoven of mechanical, thermal and chemical. The great variety of types of nonwovens according to web bonding are shown in Table 1 [33].
Table 1. Classification based on web bonding.
|
Mechanical bonding |
Needle punch |
|
Spun laced |
|
|
Stitch bonded |
|
|
Thermal bonding |
Calendaring |
|
Through air bonding |
|
|
Sonic bonding |
|
|
Chemical bonding |
Impregnating |
|
Foam coating |
|
|
Spraying |
|
|
Print bonding |
- Are the figures in this manuscript from the archive of the authors? Or are they taken from other articles? Please indicate this under the articles and provide a reference if appropriate.
All figures in the article were prepared by the authors.
- In subsection 4.2, the study shows various biodegradability of nonwovens has no citation. Is this study of the authors? Please, clarify whether it is an original article or a review; many passages look to be original articles or just the opinions of the authors?
The data presented in subchapter 4.2 are the authors' own data. The data comes from the conducted research.
- Please, use the internationally used units, replace jtk with CFU.
jtk was replaced with CFU.

Round 2
Reviewer 1 Report
The authors have revised most of the specific changes suggested. The authors should address the remaining comments before acceptance for publication.
- Line 241: it should be “the active substances can influence by”. This is raised in the first review but it has not been corrected.
- Fig 1: Are the presented data unpublished? Please check with the journal guideline as it is uncommon to include unpublished data in a review article. For the other figures, they should have proper citation in the figure caption and with copyright permission.
- For the benefit of the readers, the authors are strongly encouraged to go over the manuscript and revise for any grammatical errors and improve the style of presentation.
Author Response
Response to reviewer 1
Replies to comments and suggestions
English language and style
Moderate English changes required
English language was edited for proper grammar and overall style by native speakers.
Comments and Suggestions for Authors:
- Line 241: it should be “the active substances can influence by”. This is raised in the first review but it has not been corrected.English language was edited for proper grammar and overall style.
We agree with the reviewer's remark. Changes made as suggested.
- Fig 1: Are the presented data unpublished? Please check with the journal guideline as it is uncommon to include unpublished data in a review article. For the other figures, they should have proper citation in the figure caption and with copyright permission.
- The data presented in Figure 1 are the authors' own data. The data was obtained from research carried out by the coauthors as a result of implemented projects. All the figures and schemes have been prepared by the authors, so in our opinion there is no need to include references to the literature. In the text, a clarifying entry was added: The SEM images, were obtained on the basis of own studies, on activated biodegradable nonwovens produced in the spun-bonded technique are presented in Figure 2.
- For the benefit of the readers, the authors are strongly encouraged to go over the manuscript and revise for any grammatical errors and improve the style of presentation.
Thank you for your suggestions, the manuscript has been verified in English by a native speaker.

Reviewer 3 Report
all ambiguities were answered and all comments were responded to by the authors, I recommend publishing in the current form.
Author Response
Dear Reviewer,
Thank you for your recommendations for publication.